# Immunosenescence in Aging-Related Vascular Dysfunction

**DOI:** 10.3390/ijms232113269

**Published:** 2022-10-31

**Authors:** Anna Tylutka, Barbara Morawin, Edyta Wawrzyniak-Gramacka, Eryk Wacka, Wiktoria Nowicka, Jaroslaw Hiczkiewicz, Agnieszka Zembron-Lacny

**Affiliations:** 1Department of Applied and Clinical Physiology, Collegium Medicum University of Zielona Gora, 65-417 Zielona Gora, Poland; 2Student Research Group, Collegium Medicum University of Zielona Gora, 65-417 Zielona Gora, Poland; 3Department of Interventional Cardiology, Collegium Medicum University of Zielona Gora, 65-417 Zielona Gora, Poland

**Keywords:** endothelial progenitor cells, flow-cytometry, immune risk profile, inflammation, oxidative stress

## Abstract

The immunosenescence-related disproportion in T lymphocytes may have important consequences for endothelial dysfunction, which is a key event in vascular aging. The study was designed to assess the prognostic values of the inflammatory-immune profile to better predict and prevent vascular diseases associated with old age. Eighty individuals aged 70.9 ± 5.3 years were allocated to a low- (LGI) or high-grade inflammation (HGI) group based on CRP (<3 or ≥3 mg/L) as a conventional risk marker of cardiovascular diseases. Significant changes in inflammatory and endothelium-specific variables IL-1β, IL-6, TNFα, oxLDL, H_2_O_2_, NO, 3-nitrotyrosine, and endothelial progenitor cells (OR 7.61, 95% CI 2.56–29.05, *p* < 0.0001), confirmed their interplay in vascular inflammation. The flow-cytometry analysis demonstrated a high disproportion in T lymphocytes CD4^+^ and CD8^+^ between LGI and HGI groups. CRP was <3 mg/mL for the CD4/CD8 ratio within the reference values ≥ 1 or ≤2.5, unlike for the CD4/CD8 ratio < 1 and >2.5. The odds ratios for the distribution of CD4^+^ (OR 5.98, 95% CI 0.001–0.008, *p* < 0.001), CD8^+^ (OR 0.23, 95% CI 0.08–0.59, *p* < 0.01), and CD8CD45RO^+^ T naïve cells (OR 0.27, 95% CI 0.097–0.695, *p* < 0.01) and CD4/CD8 (OR 5.69, 95% CI 2.07–17.32, *p* < 0.001) indicated a potential diagnostic value of T lymphocytes for clinical prognosis in aging-related vascular dysfunction.

## 1. Introduction

Aging of the vasculature plays a central role in the morbidity and mortality of older adults. Recent data from in vitro and clinical studies have shown that the immune system influences endothelial dysfunction contributing to both microvascular (diabetic nephropathy, neuropathy, and retinopathy) and macrovascular (coronary artery disease, peripheral arterial disease, and stroke) diseases associated with old age [1,2]. Increased production of reactive oxygen species (ROS) by NADPH and mitochondria contributes to endothelial dysfunction and stiffening of large flexible arteries with age, as has been demonstrated in both laboratory animals and humans. There is strong evidence that endothelial dysfunction caused by increased oxidative stress contributes significantly to impaired dilation of coronary arteries, promoting myocardial ischemia and neurovascular uncoupling, impairing the moment-to-moment adjustment of cerebral blood flow to increased oxygen and nutrient demand that occurs with neuronal activation [3]. Furthermore, aging is related to a chronic low-grade inflammatory state in which macrophages, neutrophils, natural killer cells, and T and B lymphocytes act as major effectors of the immune-mediated cell responses [4]. This inflammatory state can result from the establishment of chronic oxidative stress by the immune system activity. Indeed, both oxidative stress and inflammation occur in endothelial dysfunction since excessive or uncontrolled reactive oxygen species (ROS) production can induce an inflammatory-immune response [5,6]. Therefore, the analysis of the oxi-inflammatory profile can provide more profound knowledge of the most relevant processes involved in vascular aging.

There is growing evidence that ROS generation is an important mechanism underlying T cell activation in the context of endothelial metabolism. The ROS-related control of T lymphocytes is essential to prevent dysregulated inflammation, which has been observed in cardiovascular diseases such as hypertension. The role of the adaptive immune system in hypertension has been extensively investigated, but the understanding of how the redox environments control T lymphocytes in this disease remains unclear [7]. Jackson et al. [8] provided early evidence that phagocytic NADPH oxidase was activated upon T cell receptor (TCR) stimulation, and the generation of an appropriate level of ROS in T cells from both NADPH oxidase activity as well as mitochondrial oxidative phosphorylation enhanced, sustained, and regulated TCR signaling to promote the metabolic rewiring necessary for the controlled switch between the quiescent and effector states. However, age-related changes in the antioxidant defense system and excessive ROS production result in aberrant T cell activation, differentiation, and proliferation [9,10]. T cell immunometabolism undergoes dramatic remodeling, which can lead to pathological conditions such as hypertension [11]. Guzik et al. [12] discovered for the first time that T-lymphocytes were an important determinant of hypertension as an immunological disease and supported the role of inflammation and oxidative stress in the basis of this prevalent disease. The blockade of tumor necrosis factor-alpha α (TNFα) normalized blood pressure and vascular ROS production in angiotensin II-infused animals. This study identified a previously unknown role of T lymphocytes in the development of hypertension and related vascular abnormalities [12].

The T lymphocyte pool includes subpopulations of antigen-inexperienced naïve T lymphocytes and antigen-experienced memory T lymphocytes. The human immune compartment is composed of ~10^12^ T lymphocytes in total, ~10^11^ of which are naïve [13]. Aging affects naïve CD4^+^T and CD8^+^T lymphocyte counts in a different way. The number of CD4^+^ naïve T lymphocytes is stable for most of the lifespan, but it markedly declines at the age of 70 years. Contrastingly, CD8^+^ naïve T lymphocytes appear to be more susceptible to apoptosis and thus more sensitive to age-related changes [14]. The immunosenescence-related disproportion in T lymphocytes may have important consequences for endothelial dysfunction, which is a key event in vascular aging leading to the initiation, progress, and advancement of cardiovascular diseases. According to Avinas et al. [15], the development and progression of hypertension and cardiovascular diseases could be predicted by simply detecting the total number of peripheral CD4^+^ T and CD8^+^ T cells and their ratio. Recently, we evaluated immunosenescence patterns by flow cytometry of naïve and memory T cell subpopulations, and the immune risk profile (IRP), expressed as the CD4/CD8 ratio and IgG CMV, was observed to be related to comorbidities. We demonstrated that the percentage of CD4^+^ T lymphocytes was significantly higher in hypertensive older adults, independently of CMV infections, with approximately 34% having CD4/CD8 > 2.5, and only 4% of the older adults with hypertension having CD4/CD8 < 1 [14]. The CD4/CD8 ratio could be an important marker of a healthy vascular system in older age [14]. On the basis of the gathered data on the immunosenescence-related disproportion in T lymphocytes and its complications, including the changes in endothelium metabolism from subclinical dysfunction to manifested disease, the study was designed to assess the prognostic values of the immune-inflammatory profile to better predict and prevent vascular contributions to the pathogenesis of multiple diseases associated with old age.

## 2. Results

### 2.1. Basic Characteristics of Study Sample

The body mass index (BMI) ranged from 20.7 to 32.5 kg/m^2^. Approximately 39% of the study seniors had normal body mass index (18.5–24.9 kg/m^2^), and 61% were classified as overweight and obese (≥25 kg/m^2^), mainly among individuals from the HGI group (Table 1). High-fat mass dominated among women (women 23.6 ± 5.4 kg, men 20.4 ± 5.4 kg), whereas high fat-free mass was characteristic of men (women 41.1 ± 6.6 kg, men 55.9 ± 10.2 kg). There were no significant differences in components of body composition and visceral fat content between LGI and HGI groups (Table 1). Fat mass and fat-free mass were not related to changes in circulating inflammatory biomarkers. Systolic blood pressure > 140 mmHg was recorded in 27%, and diastolic blood pressure < 70 mmHg was found in 19% of the study subjects, which increases cardiovascular risk in older patients according to the latest guidelines of the European Society of Hypertension and the European Society of Cardiology [16].

### 2.2. Hematological Variables

The white blood cell count fell within the referential range in all participants. Neutrophils, monocytes, and platelets tended toward high values in HGI, whereas lymphocyte count was significantly higher in HGI compared to the LGI group (Table 2). MPV, as a marker of platelet activation, was below the referential values in 99% of the study sample and tended toward low values by CRP ≥ 3 mg/L, which plays a role in the potentiation of thrombin-induced platelet activation. The parameters of the red blood cells also fell within the referential range except for HB concentration, which was below the referential values in 21% of the study sample, which confirms the existence of anemia in old age [17]. However, the observed low HB level did not relate to inflammatory status.

### 2.3. Lipoprotein-Lipid Profile

The total cholesterol and lipoproteins have been proven to be the strongest biomarkers of vascular aging and nutritional status [18]. In this study, high levels of TG > 150 mg/dL were found in 32%, whereas high levels of TC > 200 mg/dL and non-HDL > 130 mg/dL were found in 80% of our study sample (Table 3). However, the lipoprotein-lipid profile did not differ between LGI and HGI, contrary to oxLDL, which was two-fold higher in the HGI group. OxLDL was highly correlated with endothelium-specific variables, such as H_2_O_2_ (r_s_ = 0.621, *p* < 0.001), NO (r_s_ = 0.606, *p* < 0.001), 3-NitroT (r_s_ = 0.579, *p* < 0.001), and EPC (r_s_ = 0.600, *p* < 0.001). There is a large body of evidence that oxLDL could act on endothelial cell activities in almost every aspect, such as proliferation, differentiation, apoptosis, mobilization, migration, and senescence [19].

### 2.4. Inflammatory and Endothelium-Specific Variables

Most observational studies and clinical trials have used high-sensitivity CRP as a biochemical marker of inflammation because it is relatively stable and easy to measure. In our study, CRP concentration was found to fall within normal ranges in 44 individuals (1.276 ± 0.782 mg/L) and above normal ranges in 36 individuals (5.68 ± 3.13 mg/L); the differences were statistically significant (*p* < 0.001). CRP highly correlated with other markers of inflammaging, i.e., IL-1β (r_s_ = 0.832, *p* < 0.001) and TNFα (r_s_ = 0.771, *p* < 0.001), which were significantly higher in the HGI group (Table 4). Furthermore, CRP concentration significantly (*p* < 0.001) impacts the immune risk profile (IRP) expressed as the ratio of total CD4^+^ to total CD8^+^. The older adults with CD4/CD8 < 1 and CD4/CD8 > 2.5 demonstrated CRP of 5.50 ± 5.45 mg/L and 4.35 ± 2.77 mg/L, respectively. Meanwhile, the older adults with CD4/CD8 ≥ 1 or ≤ 2.5 had CRP 2.50 ± 2.72 mg/L (Figure 1). The classical biomarker of aging IL-6, called “a cytokine for gerontologists” [21], only tended toward higher values in the HGI group but significantly correlated with endothelium-specific variables such as oxLDL (r_s_ = 0.653, *p* < 0.001), NO (r_s_ = 0.546, *p* < 0.001), and 3NitroT (r_s_ = 0.579, *p* < 0.001). H_2_O_2_, a promoter of inflammation in endothelial and smooth muscle cells [3], showed growth in the HGI group and a high correlation with NO (r_s_ = 0.494, *p* < 0.001) and 3NitroT (r_s_ = 0.524, *p* < 0.001). Although NO generation was enhanced in HGI, its bioavailability was reduced, which was demonstrated by increased 3-NitroT concentration. The relation of 3-NitroT/NO (r_s_ = 0.725, *p* < 0.001) and 3-NitroT/oxLDL (r_s_ = 0.579, *p* < 0.001) confirmed the previous recommendation of Bencsik et al. [22] that 3-NitroT could be a new risk factor of endothelial dysfunction. The optimal threshold values (cut-offs) of inflammatory and endothelium-specific variables for clinical stratification were 2.0 pg/mL for IL-1β, 58 ng/mL for IL-6, 98.3 ng/L for TNFα, 369 ng/mL H_2_O_2_, 551.5 μmol/L for NO, and 2.3 nmol/mL for 3NitroT. These results indicate that the risk of endothelial dysfunction significantly increases above the cut-off values presented in Table 5 and Table 6. IL-1β, TNFα, NO, and 3NitroT demonstrated the highest specificity, approx. 100% in the univariate logistic model (Table 5). Substantial changes were observed for the circulating of EPC, which was two-fold higher in HGI compared to the LGI group. The AUC content (0.837) with a cut-off at 10.05 ng/mL for EPC (OR 7.61, 95% CI 2.56–29.05, *p* < 0.0001) indicates a high diagnostic value of EPC in the assessment of the course of endothelial regeneration.

### 2.5. Flow Cytometry Analysis

The percentage of CD4^+^T lymphocytes was significantly higher, whereas the CD8^+^ T lymphocyte number was lower in HGI compared to LGI (Table 7). Eventually, the immune risk profile, expressed as the CD4/CD8 ratio, was observed to be increased significantly in the HGI group. Most individuals from the LGI group (79.6%) demonstrated the CD4/CD8 ratio within the reference range ≥ 1 or ≤2.5, whereas 6.8% had CD4/CD8 < 1, and 13.6% had CD4/CD8 > 2.5 (Figure 2A). However, 41.7% of individuals from HGI demonstrated the CD4/CD8 ratio within the reference range ≥ 1 or ≤2.5, whereas 5.5% had CD4/CD8 < 1, and 52.8% had CD4/CD8 > 2.5 (Figure 2B). This means that the CD4/CD8 ratio > 2.5 is predominant in old age and closely related to inflammatory state. There were no significant differences in CD4^+^ naïve and CD4^+^ memory T lymphocytes, but the CD4CD45RA/CD4CD45RO ratio tended to reach higher values in the HGI group. A lower percentage of CD8^+^ naïve and CD8^+^ memory T lymphocytes was identified in the HGI group, but the difference in the CD8CD45RA/CD8CD45RO ratio was statistically insignificant. The result of the ROC curve analysis of CD4^+^ T lymphocytes and the CD4/CD8 ratio ranged between 0.6 and 0.7, indicating a relatively high diagnostic value of T lymphocytes for clinical prognosis in age-related vascular inflammation. The optimal threshold values corresponded to 44.5 for CD4^+^ and 2.4 for CD4/CD8 ratio (Table 8). The odds ratios for the distribution of CD4^+^ T lymphocytes (OR 5.98, 95% CI 0.001–0.0008, *p* < 0.001), CD8^+^ T lymphocytes (OR 0.23, 95% CI 0.08–0.59, *p* < 0.01), CD8CD45RO^+^ (OR 0.27, 95% CI 0.097–0.695, *p* < 0.01), and CD4/CD8 ratio (OR 5.69, 95% CI 2.07–17.32, *p* < 0.001) were strongly associated with endothelial dysfunction risk (Table 9). This confirms that close contact between circulating T lymphocytes and antigen-presenting endothelium may play a unique role in shaping oxi-inflammatory status within the periphery [23].

## 3. Discussion

Endothelial dysfunction is an essential factor preceding the development of cardiovascular diseases (CVDs). It is generally complicated by the simultaneous course of oxidative stress and inflammation in the senescence of vascular endothelial cells [3]. The risk factors of enhanced aging, such as low physical activity level and high visceral fat content, can induce alterations in endothelium morphology and metabolism, which contributes to arterial stiffness, atherosclerosis, hypertension, stroke, and coronary artery disease in older adults [14,24]. In this regard, aging is a major risk factor and represents a major global health challenge in the pathogenesis of CVDs [25].

The study group was stratified according to CRP (LGI < 3 mg/L or HGI ≥ 3 mg/L), which is one of the most widely studied inflammatory molecules for CVD evaluation [26,27]. Several processes are induced by CRP and include the release of chemoattractants, which, in turn, attract monocytes toward the endothelial barrier. CRP also upregulates the release of pro-inflammatory cytokines and inhibits the release of NO, which precedes the development of arteriosclerosis. The association between CRP and cardiovascular events was shown in a meta-analysis [28]. In our study, CRP concentration was recorded at ≥3 mg/L in 45% of our study individuals and highly correlated with pro-inflammatory cytokines IL-1β, IL-6, and TNFα. CRP demonstrated low values in older adults with the CD4/CD8 ratio within the reference value range ≥ 1 or ≤2.5, but the high values in the older adults with CD4/CD8 ratio < 1 and CD4/CD8 ratio > 2.5. The changes in T lymphocyte distribution support the inflammatory state, and this, in turn, is influenced by reactive oxygen species (ROS) generation in old age [29].

Chronic inflammatory state is manifested by the release of large amounts of IL-1β, IL-6, and TNFα from immune cells, and peripheral tissues have been shown to play an important role in the pathogenesis and progression of inflammaging [21]. High levels of IL-1β, together with IL-6 and TNFα, are associated with an increased risk of morbidity and mortality in older adults. In particular, cohort studies have indicated that IL-1β, IL-6, and TNFα are involved in the alteration of nutritional status, poor physical performance, loss of muscle strength, cognitive decline, and cardiological, neurological, and vascular events [30,31]. In our study, IL-1β and TNFα have reached particularly high concentrations (14-fold increase) and specificity approx. 100% in the univariate logistic model. This suggests the use of IL-1β and TNFα as an additional alternative for the decision-making process for use of therapies in select high-risk CVD patients. IL-6 expression is normally low in the absence of inflammation, but its elevated serum level is characteristic of aging and may reflect age-related pathological processes that develop over decades, even in apparently healthy subjects [32,33]. In our research, we did not observe statistical significance between the group and the level of IL-6; however, higher values of IL-6 were noted in the HGI group at 115.6 ± 74.5. The low AUC 0.555 (sensitivity 63.6% and specificity 19.4%) for IL-6 underlines the low diagnostic utility of this cytokine in our study. TNFα was also detected at elevated levels in most studies of older adult populations and was associated with frailty and reduced functional capacity, as well as increased CVD mortality [21,31,34,35]. TNFα plays a pivotal role in endothelial dysfunction through the disturbance it causes to the equilibrium between endothelial nitric oxide synthase (eNOS) and inducible nitric oxide synthase (iNOS) activities, which results in pro-apoptotic NO activity [36,37]. The excess NO reacts with the superoxide anion to produce peroxynitrite (ONOO), whose level increases 1,000,000-fold. ONOO, in turn, can “uncouple” eNOS to become a dysfunctional superoxide-generating enzyme that contributes to vascular oxidative stress. Without superoxide, the formation of ONOO by NO reaction with oxygen is minimal. NO and superoxide do not even have to be produced within the same cell to form peroxynitrite because NO can readily move through membranes and between cells [36]. TNFα-stimulated vascular dysfunction was reported to be induced by intra-arterial TNFα administration. In healthy volunteers, an acute local vascular inflammation was observed upon intra-arterial 30 min infusion of high-dose TNFα (80 or 240 ng/min). The administration of a lower TNFα dose (17 ng/min) for 60 min induced an increase in basal vascular resistance in healthy subjects, which was blocked by pre-treatment with a NO synthase inhibitor. The observed effects of TNFα were likely to be mediated by NO bioavailability [38]. In the present study, low NO bioavailability was demonstrated by a high concentration of 3NitroT that correlated with other inflammatory and endothelium-specific variables such as IL-6 and oxLDL. Protein nitration can have a remarkable impact on protein structure and function and has been found in different pathological conditions associated with oxidative stress status, such as CVDs and neurodegenerative diseases but also inflammation and aging [39,40]. Indeed, the tyrosine nitration of proteins, including fibrinogen, apolipoprotein A-1, and apolipoprotein B-100, has been found in the plasma collected from coronary artery disease patients, suggesting that changes in the function of some nitrotyrosine-modified proteins can create a pro-atherosclerotic milieu [41]. NO is the most important nitrosative compound, and its reduced bioavailability is crucial in endothelial dysfunction and in promoting arterial remodeling through several mechanisms [42]. Thus, the observed changes in NO, H_2_O_2_, and 3NitroT can be associated with endothelial dysfunction and can promote atherogenesis. The repair of injured endothelium is dependent on timely suppression and containment of inflammation; this process is accompanied by the activation of progenitor cells that restore tissue integrity [43]. EPC mobilization from the bone marrow is mainly triggered by inflammation [44]. As such, it was investigated by Morishita et al. [45], where they analyzed the pattern of EPC mobilization and its association with inflammation and nitro-oxidative stress markers in patients with CVDs. Similarly, we observed considerable changes in the number of circulating EPC in individuals with high oxi-inflammatory status.

Oxidative stress also manifests itself in the oxidation of LDL particles within the blood flow and vascular wall. OxLDL can trigger the expression of adhesion molecules on the cell surface and thus stimulate the activation of endothelial cells. These adhesion molecules mediate the rolling and adhesion of blood leukocytes that adhere to the endothelium and then, in response to chemokines, migrate into the intima. As a consequence of immune cell activation, ROS and pro-inflammatory cytokines are released [19]. OxLDL binds to lectin-like oxidized low-density lipoprotein receptor-1 (LOX-1), whose expression in endothelial cells is very low but can be rapidly induced by oxi-inflammatory stimuli, including IL-1β, IL-6, and TNFα [46,47,48]. In our study, oxLDL increased in patients with CRP ≥ 3 mg/L and was highly correlated with endothelium-specific variables, even though the total cholesterol, LDL, and HDL levels did not differ between groups. This confirms previous observations that the detection and quantification of oxLDL could be used as an early indicator for the pathological processes in vascular endothelium [27,49]. In the early stages of atherogenesis, oxLDL promotes endothelial injury, induces expression of adhesion molecules, and attracts macrophages [19]. In the subintimal space of the arterial vessel wall, scavenger receptor uptake of oxLDL by macrophages is viewed as a critical step in foam cell formation. OxLDL may also contribute to the destabilization of the growing plaque and to the formation of thrombi by inducing matrix metalloproteinases. Serum levels of oxLDL have already been shown to be elevated in patients with established coronary artery disease [50]. In the Edinburgh Artery Study, every 1 μmol/L increase in lipid peroxides was associated with a 17% increase in the risk of atherosclerosis [51]. The Bruneck prospective study followed 1510 men and women aged 40–79 years with ultrasound measures of carotid and femoral atherosclerosis and measured the concentration of antibodies against oxLDL in more than 90% of participants [52]. However, the interplay between lipid metabolism and immune response is not yet completely understood. In the last decade, the lymphocyte contribution to clinical manifestations of atherosclerosis has been highlighted, and effector T-cell responses seem to be exacerbated in hyperlipidemia [53,54].

Reactive oxygen species, such as H_2_O_2_ and NO, exert a dual action in the immune response [29]. This is especially true of T lymphocytes due to their high plasticity and diversity of cell types within this class of adaptive immune cells, combined with a multitude of extreme microenvironments [7]. It is well-known that H_2_O_2_ participates in the homing of T lymphocytes to sites of inflammation, which suggests a chemotactic and pro-inflammatory role of H_2_O_2_ [55]. ROS are capable of modulating immune metabolism, and the outcome is an altered T lymphocyte phenotype. The susceptibility of human T cells to H_2_O_2_-induced apoptosis strongly depends on the T-cell subset. In fact, 5 μM H_2_O_2_ induces apoptosis in CD45RO^+^ T memory but not in CD45RA^+^ T naïve cells via mitochondrial depolarization and caspase activation [56]. Similarly, CD4CD45RA^+^ T naïve cells are less likely to undergo cell death upon incubation with 5–20 μM H_2_O_2_ than CD4CD45RO^+^ T memory cells [57]. Interestingly, 100 μM H_2_O_2_ completely eliminates CD4^+^ T cells but has no effect on CD4^+^ T-cell blasts [58]. Endogenous NO produces the major effects on T lymphocyte function, and all three isoforms of nitric oxide synthase (NOS) have been found in T lymphocytes or T lymphocyte-derived cell lines. The use of NOS inhibition weakens the activation of CD8^+^ T lymphocytes and also Treg induction by inhibiting transforming growth factor β1 production, which would additionally support a pro-inflammatory environment [59]. According to Kesarwani et al. [55], endogenous NO sources show similar effects to exogenous NO supplementation in T lymphocyte phenotype promotion. In our study, a disproportion in T lymphocytes was related to the changes in oxi-inflammatory status. In the HGI group, more than half of the patients showed a CD4/CD8 ratio > 2.5 and decreased NO bioavailability, while in the LGI group, ~80% showed a CD4/CD8 ratio within the reference range ≥ 1 or ≤2.5 and increased NO bioavailability, which was demonstrated by changes in 3-NitroT concentration. Case et al. [10] showed that a loss of control over ROS generation led to aberrant T-cell development and function, which resulted in an increase in double-negative cells and a decrease in CD4 and CD8 single-positive cells in the thymus. For the first time in the Polish population, we determined the high AUC content (0.713) for the CD4/CD8 ratio, and the odds ratios (OR 5.69, 95% CI 2.07–17.32, *p* < 0.001) indicated the involvement of oxidative stress in the high immune risk profile. Nevertheless, due to the small number of reports, the research should be continued, increasing the size of the groups and taking into account different age groups of older individuals. Our thorough understanding of these relations, taking into account the disproportion in the CD4/CD8 ratio observed with age and the decreasing percentage of naïve T cells, could be a potential way to improve novel therapeutic strategies for treating immunological disorders [24].

## 4. Materials and Methods

### 4.1. Participants

A total of 124 older adults aged between 65 and 88 years were recruited between December 2019 and March 2020 at the University of the Third Age, which is an organization recommended to older adults to stay active by participating in educational programs. Details on subjects’ medical history and lifestyle were obtained through a health questionnaire described by Durstine and Moore [60]. The exclusion criteria included symptoms of acute infection such as fever, cough, malaise, etc., autoimmune and endocrine diseases, tumors, and renal failure. Eventually, eighty participants (females *n* = 62, males *n* = 18) aged 70.9 ± 5.3 years participated in the project. The older adults were allocated to two groups, i.e., low-grade inflammation (LGI *n* = 44) and high-grade inflammation (HGI *n* = 36) based on the measurement of CRP concentration as a conventional marker of systemic inflammation and cardiovascular diseases according to the reference values (CRP < 3 mg/L or ≥3 mg/L) described by Pearson et al. [26]. Approximately 12% of the study subjects suffered from stable coronary artery disease and took anti-thromboembolic drugs. The medications taken by the participants included antihypertensive (84%) and hypolipidemic (14%) drugs as well as anticoagulants, including anti-platelet agents (12%), but these medications had no effect on the allocation of the older adults in two groups. All the subjects were informed of the aim of the study and signed a written consent to participate in the project. The study protocol was approved by the Regional Bioethics Commission (No. 04/133/2020 and No. UZ/19/2021) in accordance with the Helsinki Declaration.

### 4.2. Body Composition

Body mass and body composition included fat-free mass (FFM), fat mass (FM), and visceral fat content (VFC), which were evaluated by a bioelectrical impedance analysis (BIA) using Tanita Body Composition Analyzer MC-980 (Tokyo, Japan) calibrated prior to each test session in accordance with the manufacturer’s guidelines. The reproducibility of estimated values using this BIA system has been reported previously [61,62]. The measurements were taken on an empty stomach (without food or drink) between 7:00 and 9:00 a.m., with empty bladder prior to blood sampling, and the recurrence of measurement was 98%. Duplicate measurements were made with the study participants standing upright, and the mean value was included in the final analysis.

### 4.3. Blood Sampling

Fasting blood samples were collected from the median cubital vein in the morning between 8.00 and 10.00 using S-Monovette tubes (Sarstedt AG and Co. KG, Nümbrecht, Germany). The whole blood samples were placed into specimen tubes containing EDTA and were immediately analyzed. For the other biochemical analyses, blood samples were centrifuged at 3000 rpm for 10 min, and aliquots of serum were stored at −80 °C.

### 4.4. Haematological Variables

Hematological parameters, including total white blood cell count (WBC), red blood cell count (RBC), platelet count, differential white cell count (neutrophils, lymphocytes, monocytes, eosinophils, and basophils), and hemoglobin concentration (HB) were determined by Sysmex XN-1000 (Sysmex Europe Gmbh, Norderstedt, Germany).

### 4.5. Lipoprotein-Lipid Profile

Serum triglycerides (TG), total cholesterol (TC), high-density lipoproteins (HDL), and low-density lipoproteins (LDL) were determined using BM200 Biomaxima (Lublin, Poland). The non-HDL cholesterol was calculated by subtracting HDL from the total cholesterol concentration. Oxidized low-density lipoprotein (oxLDL) was determined using ELISA kits from SunRed Biotechnology Company (Shanghai, China) with detection limit of 30.3 ng/mL.

### 4.6. Inflammatory and Endothelium-Specific Variables

C-reactive protein (CRP) was measured using a high-sensitivity commercial kit from DRG International (Springfield Township, Cincinnati, OH, USA) with a detection limit of 0.001 mg/L. Interleukin 1β (IL-1β), interleukin 6 (IL-6), and tumor necrosis factor α (TNFα) concentrations were determined by using ELISA kits from SunRed Biotechnology Company (Shanghai, China) with detection limits of 28.364 pg/mL, 0.857 ng/L, and 2.782 ng/L, respectively. Hydrogen peroxide (H_2_O_2_), nitric oxide (NO), and 3-nitrotyrosine (3-NitroT) were determined using ELISA kits from SunRed Biotechnology Company (Shanghai, China) with detection limits 7.778 ng/mL, 2.052 μmol/L, and 0.007 nmol/mL, respectively. Endothelial progenitor cells (EPC) were determined using ELISA kits from SunRed Biotechnology Company (Shanghai, China) with detection limit of 0.125 ng/mL. The average intra-assay coefficient of variation (intra-assay CV) for the enzyme immunoassay tests (ELISA) was <8%.

### 4.7. Flow Cytometry Analysis

The 8-parameter CyFlow Space Sorter flow cytometer SysmexPartec and CyLyse kit from SysmexPartec (Sysmex Europe Gmbh, Norderstedt, Germany) were used for the immunophenotypic analysis of T lymphocytes. A mix of fluorochrome-conjugated antibodies (CD8 APC, CD4 FITC, CD45 RA Pacific Blue ™, CD45RO PE) was added to 100 µL blood and incubated at room temperature for 15 min. Then, 100 µL of Leukocyte Fixation Reagent A was added sequentially (incubation in the dark for 10 min), and in the last step, 2.5 mL of Erythrocyte Lysis Reagent B was added (incubation in the dark for 20 min). After this time, measurements were made according to the method described by Tylutka et al. [14]. T helper lymphocytes CD4^+^ and cytotoxic lymphocytes CD8^+^ were expressed as a percent of gated lymphocytes. Memory and naïve subpopulations were gated by positive surface staining for CD45RO and CD45RA, respectively (Figure 3). The ratios of CD4CD45RA/CD4CD45RO and CD8CD45RA/CD8CD45RO were calculated according to Hang et al. [63]. The reference values for the CD4/CD8 ratio were adopted from McBride and Striker [64] and Strindhall et al. [65]. The ratios ≥ 1 or ≤2.5 are generally considered normal; nevertheless, there is quite a lot of variability due to past or present infections, age, genetics, ethnicity, and environmental exposure. The CD4/CD8 ratio < 1 (inverted) and >2.5 (high) is regarded as an immune risk phenotype and can be associated with immunosenescence and chronic inflammatory diseases [65].

### 4.8. Statistical Analysis

Statistical analyses were performed using RStudio, version 4.1.2 [66]. The variables were reported as mean values ± standard deviation (SD) or median (Me). The assumptions for the use of parametric or nonparametric tests were checked using the Shapiro–Wilk and Levene’s tests to evaluate the normality of the distributions and the homogeneity of variances, respectively. The significant differences in mean values between the groups were assessed by the one-way ANOVA. If the normality and homogeneity assumptions were violated, the Kruskal-Wallis nonparametric test was used. Spearman range correlation (r_s_ Spearman correlation coefficient) was used to investigate the relation between inflammatory and endothelium-specific variables. The predictive value of lymphocyte T was evaluated by measuring the area under the receiver operating characteristic curve (ROC curve). The optimal cut-off for clinical stratification was obtained by calculating the Youden index. A logistic regression model was applied to investigate the risk factors associated with immunosenescence. The odds ratio (OR), 95% confidence interval (CI), and *p*-value are presented.

## 5. Conclusions

Our study demonstrated that aging is associated with profound changes in the inflammatory-immune profile, pointing to a different interplay between endothelial cells, immune cells, and oxi-inflammatory mediators at an older age. A better understanding of the mechanisms underlying T lymphocyte trafficking may increase our knowledge of the senescence of vascular endothelial cells and also explain the utility of the potential diagnostic value of T lymphocytes for clinical prognosis in aging-related vascular dysfunction. There were some limitations that should be taken into account in subsequent studies, such as the effect of physical activity and nutrition on the analyzed immune cells and cytokine levels.

## Figures and Tables

**Figure 1 ijms-23-13269-f001:**
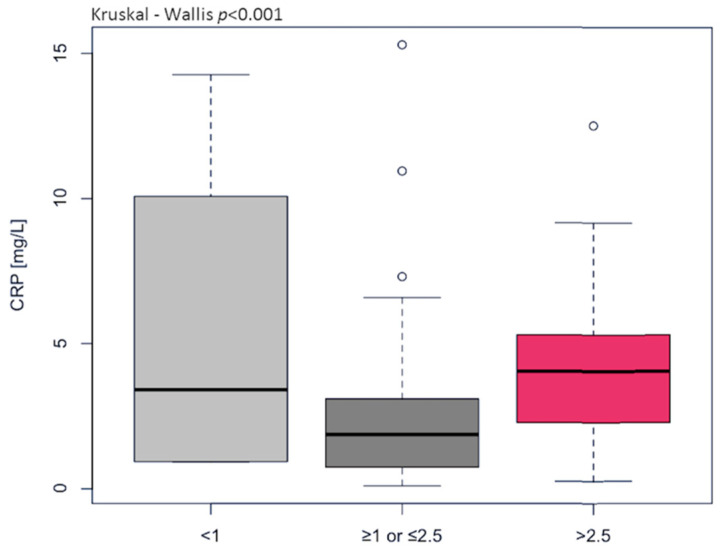
The levels of C-reactive protein (CRP) differentiated the immune risk profile (IRP) expressed as the ratio of total CD4^+^ to total CD8^+^ (normal ≥ 1 or ≤2.5, inverted < 1, high > 2.5).

**Figure 2 ijms-23-13269-f002:**
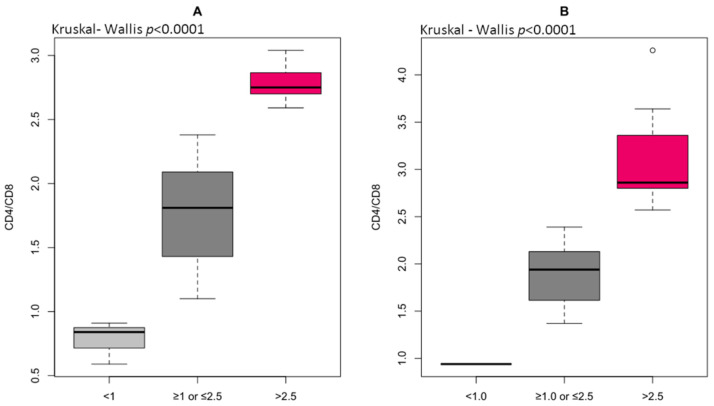
The immune risk profile (IRP) expressed as the ratio of total CD4^+^ to total CD8^+^ in low-grade inflammation group (**A**) and high-grade inflammation group (**B**) differentiated to reference ranges for CD4/CD8 (normal ≥ 1 or ≤2.5, inverted < 1, high > 2.5).

**Figure 3 ijms-23-13269-f003:**
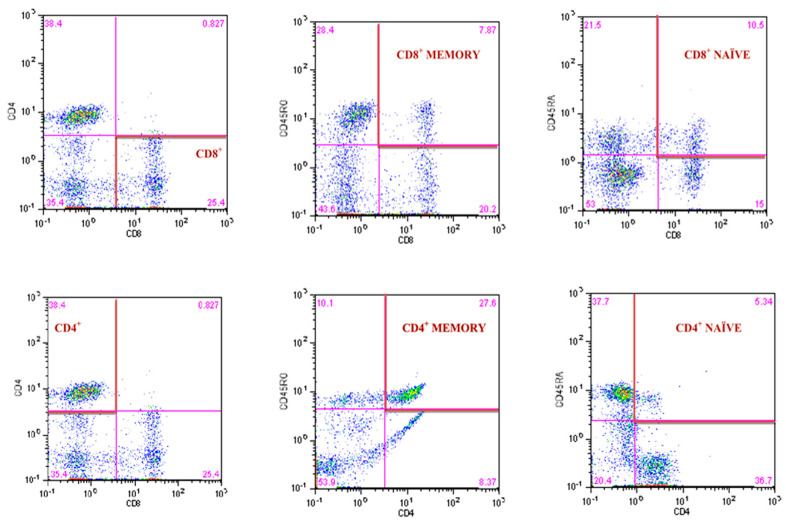
Gating strategy for identifying the CD4^+^ and CD8^+^ T lymphocyte and the frequency of CD4^+^ and CD8^+^ naïve and memory T lymphocytes.

**Table 1 ijms-23-13269-t001:** Basic characteristics of study sample.

	LGI n = 44 Mean ± SD (Me)	HGI n = 36 Mean ± SD (Me)	*p*-Value
Age (years)	70.3 ± 5.3 (71.0)	71.6 ± 5.3 (71.0)	0.281
Weight (kg)	67.1 ± 8.2 (66.1)	70.5 ± 9.7 (69.0)	0.088
Height (cm)	162.3 ± 6.2 (163.0)	160.7 ± 5.7 (160.3)	0.269
BMI (kg/m^2^)	25.4 ± 2.4 (25.0)	27.7 ± 3.2 (27.3)	<0.001
VFC (VF unit)	10.2 ± 2.8 (9.0)	11.4 ± 3.9 (10.0)	0.021
FM (kg)	21.7 ± 5.3 (21.4)	24.1 ± 5.4 (22.8)	0.051
FM%	32.8 ± 7.7 (33.6)	34.0 ± 5.3 (33.0)	0.677
FFM (kg)	44.4 ± 8.2 (43.0)	44.9 ± 10.6 (44.6)	0.381
SBP (mmHg)	129.5 ± 12.4 (130.5)	132.6 ± 16.5 (130.5)	0.340
DBP (mmHg)	79.2 ± 10.5 (78.0)	78 ± 10.1 (77.5)	0.603

LGI, low-grade inflammation group; HGI, high-grade inflammation group; BMI, body mass index; VFC, visceral fat content; FM, fat mass; FFM, fat-free mass; SBP, systolic blood pressure; DBP, diastolic blood pressure; SD, standard deviation; Me, median.

**Table 2 ijms-23-13269-t002:** Hematological variables.

	Reference Values	LGI n = 44 Mean ± SD (Me)	HGI n = 36 Mean ± SD (Me)	LGI vs. HGI *p*-Value
WBC (10^3^/µL)	4.0–10.2	5.76 ± 1.39 (5.58)	6.23 ± 1.83 (5.94)	0.333
Neutrophils (10^3^/µL)	2.0–6.9	3.17 ± 1.09 (2.98)	3.38 ± 1.28 (3.12)	0.691
Lymphocytes (10^3^/µL)	0.6–3.4	1.81 ± 0.61 (1.75)	2.05 ± 0.68 (2.02)	0.038
Monocytes (10^3^/µL)	0.00–0.90	0.50 ± 0.11 (0.49)	0.53 ± 0.20 (0.54)	0.847
Platelets (10^3^/µL)	140–420	223 ± 41 (224)	233 ± 59 (223)	0.702
MPV (fL)	8.9–11.8	7.20 ± 0.88 (7.18)	6.90 ± 0.83 (6.91)	0.129
RBC (10^3^/µL)	F 4.0–5.5 M 4.5–6.6	4.55 ± 0.29 (4.50)	4.50 ± 0.30 (4.44)	0.363
HB (g/dL)	F 12.5–16.0 M 13.5–18.0	13.83 ± 0.88 (13.85)	13.68 ± 0.94 (13.60)	0.469
HCT%	F 37.0–47.0 M 40.0–51.0	41.83 ± 2.51 (41.50)	41.29 ± 2.67 (41.15)	0.355
MCV (fL)	80.0–97.0	91.97 ± 3.58 (92.0)	91.92 ± 3.25 (91.45)	0.951
MCH (pg/RBC)	26.0 –32.0	30.50 ± 1.33 (30.60)	30.58 ± 1.16 (30.50)	0.759
MCHC (g/dL)	31.0–36.0	33.16 ± 0.57 (33.20)	33.26 ± 0.50 (33.40)	0.423
RDW%	11.5–14.8	14.89 ± 0.81 (14.95)	15.11 ± 0.82 (15.25)	0.223

LGI, low-grade inflammation group; HGI, high-grade inflammation group; WBC, white blood cells; MPV, mean platelet volume; RBC, red blood cells; HB, hemoglobin; HCT, hematocrit; MCV, mean cell volume; MCH, mean corpuscular hemoglobin; MCHC, mean corpuscular hemoglobin concentration; RDW, red cell distribution width; SD, standard deviation; Me, median.

**Table 3 ijms-23-13269-t003:** Lipoprotein-lipid profile.

	Reference Values *	LGI n = 44 Mean ± SD (Me)	HGI n = 36 Mean ± SD (Me)	LGI vs. HGI *p*-Value
TG (mg/dL)	<150	139.8 ± 19.0 (139.2)	143.7 ± 31.6 (136.5)	0.846
TC (mg/dL)	125–200	234.7 ± 35.1 (237.0)	226.4 ± 41.5 (223.1)	0.339
LDL (mg/dL)	<116	89.1 ± 26.6 (91.1)	80.9 ± 32.3 (80.4)	0.224
HDL (mg/dL)	>40	73.5 ± 15.8 (72.7)	70.3 ± 19.2 (68.0)	0.465
non-HDL (mg/dL)	<130	161.2 ± 39.1 (161.1)	156.1 ± 47.5 (153.5)	0.568
oxLDL (mg/dL)	-	0.018 ± 0.020 (0.009)	0.041 ± 0.041 (0.024)	0.020

LGI, low-grade inflammation group; HGI, high-grade inflammation group; TG, triglycerides; TC, total cholesterol; LDL, low-density lipoproteins; HDL, high-density lipoproteins, non-HDL cholesterol calculated by subtracting the HDL value from a TC; oxLDL, oxidized low-density lipoprotein; SD, standard deviation; Me, median. * According to the American College of Cardiology for patients with low CVD risk [20].

**Table 4 ijms-23-13269-t004:** Inflammatory and endothelium-specific variables.

	LGI n = 44 Mean ± SD (Me)	HGI n = 36 Mean ± SD (Me)	LGI vs. HGI *p*-Value
IL-1β (pg/mL)	0.862 ± 0.558 (0.787)	10.849 ± 8.436 (7.536)	<0.001
IL-6 (ng/mL)	105.0 ± 67.7 (68.8)	115.6 ± 74.5 (73.4)	0.400
TNFα (ng/L)	49.0 ± 21.9 (44.6)	689.0 ± 831.5 (235.6)	<0.001
H_2_O_2_ (ng/mL)	718.8 ± 703.7 (349.0)	801.2 ± 718.9 (420.5)	0.585
NO (μmol/L)	265.6 ± 128.9 (215.6)	321.2 ± 169.2 (263.4)	0.121
3NitroT (nmol/mL)	1.183 ± 0.456 (1.055)	1.510 ± 0.844 (1.081)	0.248
EPC (ng/mL)	11.39 ± 6.08 (9.86)	21.68 ± 11.90 (17.23)	<0.001

CRP, C-reactive protein; IL-1β, interleukin 1β; IL-6, interleukin 6; TNFα, tumor necrosis factor α; NO, nitric oxide; H_2_O_2_, hydrogen peroxide; 3NitroT, 3-nitrotyrosine; EPC, endothelial progenitor cells; SD, standard deviation; Me, median.

**Table 5 ijms-23-13269-t005:** The statistical characteristics of the ROC curve for the univariate logistic model for inflammatory and endothelium-specific variables.

	AUC	Cut-Off Value	Sensitivity (%)	Specificity (%)
IL-1β (pg/mL)	0.992	2.0	91.6	97.7
IL-6 (ng/mL)	0.555	58.0	63.6	19.4
TNFα (ng/L)	0.972	98.3	88.0	100.0
H_2_O_2_ (ng/mL)	0.545	369.0	60.0	56.2
NO (μmol/L)	0.601	551.5	22.2	100.0
3NitroT (nmol/mL)	0.584	2.3	22.2	100.0
EPC (ng/mL)	0.837	10.5	97.2	61.3

AUC, area under the curve, cut-off value, and the optimal threshold value for clinical stratification.

**Table 6 ijms-23-13269-t006:** The odds ratios for the distribution of inflammatory and endothelium-specific variables. The parameter ranges given in the rows of the contingency tables were determined based on the cut-off values calculated on the ROC curves.

Variables	OR	95% CI	*p*-Value
IL-1β (pg/mL)	258.1	40.98–6992.32	<0.0001
IL-6 (ng/mL)	1.62	0.623–4.34	0.320
TNFα (ng/L)	205.6	33.93–5482.33	<0.0001
H_2_O_2_ (ng/mL)	1.55	0.49–4.96	0.448
NO (μmol/L)	9.08	1.46–240.54	0.015
3NitroT (nmol/mL)	9.08	1.46–240.54	0.015
EPC (ng/mL)	7.61	2.56–29.05	<0.0001

OR, odds ratio; 95% CI, confidence interval.

**Table 7 ijms-23-13269-t007:** Distribution of T lymphocytes.

	LGI n = 44 Mean ± SD (Me)	HGI n = 36 Mean ± SD (Me)	LGI vs. HGI *p*-Value
CD4^+^	35.59 ± 9.64 (38.36)	40.50 ± 12.87 (45.00)	0.015
CD8^+^	21.43 ± 6.79 (21.40)	17.66 ± 7.33 (17.34)	0.005
CD4/CD8	1.84 ± 0.60 (1.88)	2.45 ± 0.78 (2.60)	<0.001
CD4CD45RA^+^	6.00 ± 3.88 (6.22)	6.50 ± 5.13 (4.53)	0.797
CD4CD45RO^+^	22.67 ± 7.39 (22.66)	21.88 ± 8.53 (25.24)	0.911
CD4CD45RA/CD4CD45RO	0.29 ± 0.21 (0.19)	0.34 ± 0.32 (0.31)	0.430
CD8CD45RA^+^	11.43 ± 6.90 (8.98)	8.84 ± 5.22 (8.15)	0.180
CD8CD45RO^+^	9.88 ± 5.17 (10.23)	7.05 ± 3.96 (6.00)	0.014
CD8CD45RA/CD8CD45RO	1.36 ± 1.05 (1.20)	1.47 ± 0.76 (1.27)	0.169

LGI, low-grade inflammation group; HGI, high-grade inflammation group; SD, standard deviation; Me, median.

**Table 8 ijms-23-13269-t008:** The statistical characteristics of the ROC curve for the univariate logistic model for T lymphocyte distribution.

	AUC	Cut-Off Value	Sensitivity (%)	Specificity (%)
CD4^+^	0.654	44.5	18.1	41.7
CD8^+^	0.317	18.9	68.1	69.5
CD4/CD8	0.713	2.4	16.0	44.4
CD4CD45RA^+^	0.517	2.92	83.3	36.3
CD4CD45RO^+^	0.507	27.55	47.2	72.7
CD4CD45RA/CD4CD45RO	0.556	0.12	91.6	29.5
CD8CD45RA^+^	0.413	12.0	0.0	54.5
CD8CD45RO^+^	0.341	9.83	25.0	43.2
CD8CD45RA/CD8CD45RO	0.589	1.85	38.9	79.5

AUC, area under the curve, cut-off value, and the optimal threshold value for clinical stratification.

**Table 9 ijms-23-13269-t009:** The odds ratios for the distribution of T lymphocytes. The parameter ranges given in the rows of the contingency tables were determined based on the cut-off values calculated on the ROC curves.

Variables	OR	95% CI	*p*-Value
CD4^+^	5.98	0.001–0.008	<0.0001
CD8^+^	0.23	0.08–0.59	0.002
CD4/CD8	5.69	2.07–17.32	<0.001
CD4CD45RA^+^	3.42	1.158–11.891	0.025
CD4CD45RO^+^	1.28	0.505–3.271	0.597
CD4CD45RA/CD4CD45RO	2.44	0.851–7.817	0.098
CD8CD45RA^+^	0.24	0.072–0.731	0.010
CD8CD45RO^+^	0.27	0.097–0.695	0.006
CD8CD45RA/CD8CD45RO	1.54	0.559–4.339	0.399

OR, odds ratio; 95% CI, confidence interval.

## Data Availability

The data used to support the findings of this study are available from the corresponding author upon request.

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
