# Peer review of "Immunosenescence in Aging-Related Vascular Dysfunction"

_ijms, 2022, doi:10.3390/ijms232113269_

Round 1
Reviewer 1 Report
The study by Tylutka et al. entitled, “Immunosenescence in Aging-related Vascular Dysfunction” sought to determine the relation between baseline inflammatory status (as assessed via hs-CRP) and immune cell phenotypes in older adults. The work highlighted in this study is novel in the sense that it establishes a new phenotype (CD4/CD8 T Cell ratio) that could serve a potential therapeutic target, or at least a biomarker for aging. There are currently a number of weaknesses with this manuscript (described below), that if addressed, could greatly enhance the biomedical impact of this work.
- “Elderly” or “older people” should be switched to “older adults” throughout
- Relationships only exist between people, so when referring to how your data is associated, it should be “relation”
- Representative diagrams should be provided for flow cytometry analyses
- There are no measures of senescence per se
o If there are studies establishing the CD4+/CD8+ ratio as a biomarker of senescence, that should be stated
- Much emphasis is placed on endothelial cells, but nothing directly related to endothelial cells is assessed in this study. Authors could consider how the plasma from these subjects influences nitric oxide production in endothelial cells in culture – this experiment would tie together the hypothetical endothelial cell phenotypes proposed in the manuscript with actual endothelial cell function.
- Did authors control for physical activity levels and/or habitual dietary intake in these subjects? If so, that information should be included
- Was there a hydration protocol that was followed prior to BIA measurements? Hydration status could greatly affect the measurements, so it should be considered.
- When reporting body mass index, it cannot be referred to as strictly “body mass”
- It is an overreach to refer to cytokines measured in blood as endothelial dysfunction markers
Author Response
Response to Reviewer 1
We greatly appreciate your time and effort dedicated to providing feedback on our manuscript and we are grateful for the insightful comments on and valuable improvements to our paper. All the suggestions helped us to evaluate our outcomes even more precisely in order to deliver improved, high quality scientific manuscript which we hope will now meet the high standards of International Journal of Molecular Sciences.
Comments and Suggestions for Authors
The study by Tylutka et al. entitled, “Immunosenescence in Aging-related Vascular Dysfunction” sought to determine the relation between baseline inflammatory status (as assessed via hs-CRP) and immune cell phenotypes in older adults. The work highlighted in this study is novel in the sense that it establishes a new phenotype (CD4/CD8 T Cell ratio) that could serve a potential therapeutic target, or at least a biomarker for aging. There are currently a number of weaknesses with this manuscript (described below), that if addressed, could greatly enhance the biomedical impact of this work.
“Elderly” or “older people” should be switched to “older adults” throughout
As suggested by Reviewer, "older people" and „elderly” were consistently replaced with "older adults" throughout the manuscript.
Relationships only exist between people, so when referring to how your data is associated, it should be “relation”
Thank you very much for pointing this out. Consequently, the term "relationship" has been changed to "relation" throughout the manuscript.
Representative diagrams should be provided for flow cytometry analyses
Following the Reviewer’s suggestion Figure 2 which includes the gating strategy has been added.
There are no measures of senescence per se o If there are studies establishing the CD4+/CD8+ ratio as a biomarker of senescence, that should be stated
During the past decade, three prospective cohort studies with the participation of Swedes, Dutch and Belgians were performed to assess the immune risk profile (IRP) in the elderly defined by, among others, CD4/CD8 ratio [Adriaensen et al. 2012]. The inconsistencies may be ascribed to a large number of factors, including gender, age, nutrition, amount of physical activity or fat content, which can all affect the ratio, and also to the fact that the values ³1 or ≤2.5 are commonly used as the reference values [McBride JA and Striker R 2017]. Neither are they unanimous as to whether the rise or the decline in the ratio is more favourable to maintain the longevity of the elderly. The CD4/CD8 ratio was found to increase with age in OCTO/NONA surviving participants over 100 years of age [Strindhal et al. 2012]. On the other hand, the analysis by Vasson et al. [2013] showed a decreasing trend of the CD4/CD8 with age in Spanish and French population. In our published study [Tylutka et al. 2021] we searched for the answer whether lifestyle exercise had an effect on the CD4/CD8 ratio. Interestingly, the CD4/CD8 ratio was found to fall within the range of the reference values in 55.9% of the group of older active participants. Our study group of active older adults was classified as the group representing healthy ageing. The CD4/CD8 ratio is also contingent on the body fat content and in all our study participants (both active and inactive), high fat content shifted the CD4/CD8 ratio <1. The immune risk profile, which would assess the potential rate of aging, in addition to the CD4CD8 ratio, also assesses other changes such as cytomegalovirus seropositivity, reduction in naïve T count and others. In our study, we assessed only two of the aging indicators included in the IRP in relation to aging-related vascular dysfunction.
Much emphasis is placed on endothelial cells, but nothing directly related to endothelial cells is assessed in this study. Authors could consider how the plasma from these subjects influences nitric oxide production in endothelial cells in culture – this experiment would tie together the hypothetical endothelial cell phenotypes proposed in the manuscript with actual endothelial cell function.
Thank you very much for this remark. We absolutely agree with the Reviewer. In subsequent studies we will include the analysis of the influence of nitric oxide production in endothelial cells in culture.
Did authors control for physical activity levels and/or habitual dietary intake in these subjects? If so, that information should be included
In the conclusion, the following limitation section has been added: „There were some limitations that should be taken into account in subsequent studies, such as the effect of physical activity and nutrition on the analyzed immune cells and cytokine levels”.
Was there a hydration protocol that was followed prior to BIA measurements? Hydration status could greatly affect the measurements, so it should be considered.
Before the body composition analysis, the patients were informed about how to prepare. The tests were carried out between 7:00 am and 9:00 am before blood sampling. The patients to be examined with an empty bladder and an empty stomach i.e., they did not eat or drink 12 hours before the examination. The relevant changes have been made to section 2.2.
When reporting body mass index, it cannot be referred to as strictly “body mass”
I absolutely agree with the Reviewer. This entry in Section 3.1 has been corrected
It is an overreach to refer to cytokines measured in blood as endothelial dysfunction
Please forgive us the verbal awkwardness, but we were prompted to use it by earlier publications. Specifically, endothelial dysfunction is associated with reduced NO production, anticoagulant properties, increased platelet aggregation, increased expression of adhesion molecules, increased expression of chemokines and cytokines, and increased reactive oxygen species production from the endothelium [Al-Isa AN et al. Atherosclerosis 2010 doi: 10.1016/j.atherosclerosis.2009.07.056, Zhang H et al. Clin Sci (Lond) 2009 https://doi.org/10.1042/CS20080196]. These molecules play important roles in the development of vascular complications including atherosclerosis and other vascular pathologies. Importantly, endothelial dysfunction has been shown to be of prognostic significance in predicting vascular events [Corrado E. et al. Coronary Artery Disease 2008 doi: 10.1097/MCA.0b013e3282f3fbde, Perticone F et al. Circulation 2008 doi: 10.1161/01.cir.104.2.191], so endothelial function testing may potentiate the detection of cardiovascular diseases such as myocardial infarction, peripheral vascular disease, ischemic stroke, and others [Suuronen EJ et al. J Thorac Cardiovasc Surg 2010 doi: 10.1016/j.jtcvs.2009.04.055, Kolluru GK et al. Int J Vasc Med 2012 doi: 10.1155/2012/918267].

Reviewer 2 Report
This is an original research article, which aims to assess the prognostic values of oxidative, inflammatory and immune markers associated with older age, so as to predict and prevent cardiovascular disease.
Generally, the topic is quite interesting, and the authors have in depth knowledge. They have used the appropriate methodology, study design, and adequate statistical analysis. The results are sufficiently well-presented, clear, and easy to understand, so as to reach safe and solid conclusions. Overall, the manuscript is well written and structured. Thus, I think it would make a nice addition to International Journal of Molecular Sciences as an original research article.
However, the following points should be generally considered, thus minor revision is demanded.
1) The abstract should be modified as following; background, aim, methods, results, conclusion.
2) Further analysis of the pathophysiology of vascular aging would be a nice addition in the part of introduction.
3) Kindly refer the comorbidities and contaminant medication among each group, if available.
4) Results of IL-6 were deprived of statistical significance, although there was a tendency. Kindly further discuss.
5) Use the same abbreviations such as CVDs throughout the whole manuscript.
6) Line 330-332; Ref 34 reported a different cut-off value of CRP (>5mg/L) among patients with PAD, which differentiates from your presented results. Kindly comment.
7) Ref 39 includes Sicilian population. Can this statement be applied to general population?
8) Ref 67 does not exist. Kindly correct.
Author Response
Response to Reviewer 2
We greatly appreciate your time and effort dedicated to providing feedback on our manuscript and we are grateful for the insightful comments on and valuable improvements to our paper. All the suggestions helped us to evaluate our outcomes even more precisely in order to deliver improved, high quality scientific manuscript which we hope will now meet the high standards of International Journal of Molecular Sciences.
Comments and Suggestions for Authors
This is an original research article, which aims to assess the prognostic values of oxidative, inflammatory and immune markers associated with older age, so as to predict and prevent cardiovascular disease. Generally, the topic is quite interesting, and the authors have in depth knowledge. They have used the appropriate methodology, study design, and adequate statistical analysis. The results are sufficiently well-presented, clear, and easy to understand, so as to reach safe and solid conclusions. Overall, the manuscript is well written and structured. Thus, I think it would make a nice addition to International Journal of Molecular Sciences as an original research article. However, the following points should be generally considered, thus minor revision is demanded.
1) The abstract should be modified as following; background, aim, methods, results, conclusion.
The abstract has been modified accordingly.
2) Further analysis of the pathophysiology of vascular aging would be a nice addition in the part of introduction.
Following the Reviewer’s suggestion, the introduction section has been improved.
3) Kindly refer the comorbidities and contaminant medication among each group, if available.
Section 2.1. Participants has been enriched with the following information: “Approximately 12% of the study subjects suffered from a stable coronary artery disease and took anti-thromboembolic drugs. The medications taken by the control group included antihypertensive (84%) and hypolipidemic (14%) drugs as well as anticoagulants including anti-platelet agents (12%), but these medications had no effect on the allocation of the older adults in two groups.”
4) Results of IL-6 were deprived of statistical significance, although there was a tendency. Kindly further discuss.
Thank you very much for this remark. Interleukin-6 is usually present at low levels in the blood, but its increase is associated with aging and with other diseases that may be associated with low-grade chronic inflammation. Consequently, we cannot evaluate the cytokine as a marker only for cardiovascular disease or endothelial dysfunction. In our study, we did not observe statistically significant differences in the level of IL-6 between the groups (p = 0.400), but a higher value of IL-6 was noted in HGI group. The description of IL-6 was modified accordingly in the discussion section.
5) Use the same abbreviations such as CVDs throughout the whole manuscript.
Thank you very much for pointing this out. Consequently, the term „CVDs” has been used throughout the manuscript.
6) Line 330-332; Ref 34 reported a different cut-off value of CRP (>5mg/L) among patients with PAD, which differentiates from your presented results. Kindly comment.
Thank you very much for this comment. A large number of studies focus on the evaluation of the acute phase protein (CRP) as one of the inflammatory molecules for CVDs. It should be emphasized here that CRP is not a specific protein, and an increase in its concentration can be observed in any disease accompanied by inflammation. In the case of endothelial disfunction or PAD, the increase in CRP protein may be different, depending on the stage of the disease, the age of the patient or co-morbidities. In the Assessment of Serum Neopterin as a Biomarker in Peripheral Artery Disease study, we assessed circulating NPT and its interaction with conventional endothelial dysfunction markers and vascular regenerative potential in a patient population with peripheral artery diseases who were stratified by ABI. Our study included 59 middle-aged people and the following inclusion criteria were applied: age >60 years, documented, stable, at least 3-month intermittent claudication, and ABI ≤ 0.9. A total of 60 patients, females (n = 42) and males (n = 18), with no history of PAD and ABI> 0.9 constituted a reference group. CRP concentration > 5 mg /L was found in 30% of the patients with ABI ≤ 0.9 and in 100% of the patients with ABI ≤ 0.5. NPT concentration reached the values> 10 nmol / L in the same group of patients, indicating the association of NPT with CRP as a circulating biomarker in PAD severity. NPT concentration highly correlated with CRP and ABI. In the manuscript submitted for evaluation in IJMS, the criterion for the division of the study group (stratification) was based on reference values (CRP <3 mg/L or ≥3 mg /L) described by Pearson et al. [2003] which suggested that the measurement of CRP concentration was a conventional marker of systemic inflammation and cardiovascular diseases.
7) Ref 39 includes Sicilian population. Can this statement be applied to general population?
Aging is characterized by a remodeling process of immune system where several functions are reduced, whereas others remain unchanged or increased. The important hallmarks of aging includea persistence of a low-grade chronic inflammatory status and a consistent increase of activated cells which progressively fill the immunological space [Palmeri et al. 2012]. The changes in the level of cytokines observed with age, e.g. an increase in IL-6, TNF alpha, do not only depend on the studied population (their age), but also on associated comorbidities. It is generally accepted that the onset of low-grade inflammation leads to changes in the immune cell subpopulation, although these changes may vary from patient to patient.
8) Ref 67 does not exist. Kindly correct.
References section has been corrected.

Round 2
Reviewer 1 Report
Thank you for taking the time to thoughtfully respond to my comments. I believe the manuscript has been markedly improved.